# A Rare Variant in *MDH2* (rs111879470) Is Associated with Predisposition to Recurrent Breast Cancer in an Extended High-Risk Pedigree

**DOI:** 10.3390/cancers15245851

**Published:** 2023-12-15

**Authors:** Lisa A. Cannon-Albright, Jeff Stevens, Craig C. Teerlink, Julio C. Facelli, Kristina Allen-Brady, Alana L. Welm

**Affiliations:** 1Genetic Epidemiology Group, Department of Internal Medicine, University of Utah School of Medicine, Salt Lake City, UT 84132, USAcraig.teerlink@utah.edu (C.C.T.); kristina.allen@utah.edu (K.A.-B.); 2George E. Wahlen Department of Veterans Affairs Medical Center, Salt Lake City, UT 84148, USA; 3Huntsman Cancer Institute, Salt Lake City, UT 84132, USA; alana.welm@hci.utah.edu; 4Department of Biomedical Informatics and Utah Clinical and Translational Science Institute, University of Utah School of Medicine, Salt Lake City, UT 84132, USA; julio.facelli@utah.edu; 5Department of Oncological Sciences, University of Utah School of Medicine, Salt Lake City, UT 84132, USA

**Keywords:** recurrent breast cancer, predisposition, pedigree, UPDB, *CHEK2*, *PMS2*, *MDH2*

## Abstract

**Simple Summary:**

The genetic variants responsible for lethal recurrent breast cancer are not yet recognized. Five sets of cousins with recurrent breast cancer who belonged to high-risk pedigrees were sequenced to identify rare, shared candidate predisposition variants in the pedigrees. The candidates were tested for association with breast cancer risk in other populations, and additional breast cancer cases were assayed for some of the candidate variants to test for co-segregation of the variants in pedigrees. One hundred and eighty-one rare candidate predisposition variants were shared in at least one cousin pair. A rare variant in *MDH2* was found to segregate with breast-cancer-affected relatives in one extended pedigree. This small sequencing study identified a set of strong candidate variants for inherited predisposition for breast cancer recurrence, including *MDH2*, which should be pursued in other resources.

**Abstract:**

A significant fraction of breast cancer recurs, with lethal outcome, but specific genetic variants responsible have yet to be identified. Five cousin pairs with recurrent breast cancer from pedigrees with a statistical excess of recurrent breast cancer were sequenced to identify rare, shared candidate predisposition variants. The candidates were tested for association with breast cancer risk with UKBiobank data. Additional breast cancer cases were assayed for a subset of candidate variants to test for co-segregation. Three-dimensional protein structure prediction methods were used to investigate how the mutation under consideration is predicted to change structural and electrostatic properties in the mutated protein. One hundred and eighty-one rare candidate predisposition variants were shared in at least one cousin pair from a high-risk pedigree. A rare variant in *MDH2* was found to segregate with breast-cancer-affected relatives in one extended pedigree. *MDH2* is an estrogen-stimulated gene encoding the protein malate dehydrogenase, which catalyzes the reversible oxidation of malate to oxaloacetate. The molecular simulation results strongly suggest that the mutation changes the NAD^+^ binding pocket electrostatics of *MDH2*. This small sequencing study, using a powerful approach based on recurrent breast cancer cases from high-risk pedigrees, identified a set of strong candidate variants for inherited predisposition for breast cancer recurrence, including *MDH2*, which should be pursued in other resources.

## 1. Introduction

Twenty to thirty percent of breast cancer recurs after initial treatment. Although recurrence is usually lethal, it is not yet possible to predict those breast cancers most likely to recur. While most studies have focused on genetic changes in breast tumors, the focus here is on identifying inherited variants that segregate in pedigrees that have an excess of recurrent breast cancer cases. This high-risk pedigree approach used a unique Utah resource that combines the genealogy of the state with decades of statewide cancer and death certificate data and the Genetic Epidemiology biorepository. Multiple candidate variants were tested for association with breast cancer risk and evaluated for co-segregation with breast cancer.

## 2. Data/Methods

The Genetic Epidemiology group at the University of Utah has recruited high-risk cancer pedigrees since the 1970s. The Genetic Epidemiology biorepository includes stored germline DNA, extracted from whole blood, for ~35,000 individuals who have deep known genealogy and are members of pedigrees studied for an excess of different cancers, including breast cancer, colorectal cancer, prostate cancer, and melanoma. High-risk pedigree studies remain a powerful mechanism for identification of predisposition genes and variants [1,2,3,4]. Here, we took advantage of unique Utah resources, combined with an unusual and powerful study design that includes sampled affected cousin pairs, to generate, and begin to evaluate, a strong list of candidate predisposition variants for recurrent breast cancer. The strength of our high-risk pedigree approach depends on the hypothesis that the related recurrent breast cancer cases are members of pedigrees with a significant excess of recurrent breast cancer cases and, thus, are likely to share an inherited predisposition from a common ancestor. Since cousins are only expected to share 1/8 of their genetic material, and these cousins belong to validated high-risk breast cancer pedigrees, any rare variants shared in these affected cousin pairs likely represent strong candidate predisposition variants. Analysis of high-risk cancer pedigrees identified in this powerful Utah resource previously provided the identification of *BRCA1* and *BRCA2*, as well as *CDKN2A* [5,6,7], which remain the most common cancer predisposition genes to be identified; the resource has identified numerous other predisposition variants for other cancers.

Utah Population Database: The Utah Population Database (UBDB) combines the genealogies of the original Utah founders from the mid-1800s and their descendants with statewide cancer and death certificate data. Almost 3 million individuals are part of at least 3 generations of genealogy, and their pedigrees can extend to 12 generations. Death certificate data for the state of Utah from 1904 with all causes of death coded in Revisions 6 to 10 of the International Classification of Diseases coding (ICD) have been record-linked to the Utah genealogy; 510,739 individual death certificates link to an individual with at least three generations of genealogy. The Utah Cancer Registry (UCR), established in 1966, records all independent primary cancers diagnosed or treated in Utah. The UCR became an NCI Surveillance, Epidemiology, and End-Results (SEER) registry in 1973. A total of 148,886 individuals with a diagnosis of cancer in the UCR are linked to an individual with at least three generations of genealogy in UPDB; 28,511 of these individuals had a breast cancer diagnosis in the UCR.

Recurrent breast cancer cases: The UCR does not record information on recurrence of cancer; all breast cancer diagnoses in the UCR represent independent primary cancers. Individuals whose breast cancer recurred were defined within the UPDB as females with a UCR diagnosis of primary breast cancer, who also had a linked Utah death certificate that included breast cancer as a cause of death, with death occurring at least 10 years after the original primary breast cancer diagnosis. The Genetic Epidemiology Biorepository stores DNA samples extracted from over 1000 breast cancer cases and over 6500 of their relatives in over 500 high-risk breast cancer pedigrees; 129 of these stored DNAs belong to individuals classified here as recurrent breast cancer cases.

High-risk recurrent breast cancer pedigrees: All of the ancestors of the 129 sampled individuals identified as recurrent breast cancer cases in the biorepository were analyzed in order to identify all clusters (pedigrees) including at least two sampled recurrent breast cancer cases sharing a common ancestor. To identify which of these pedigrees exhibited an excess of recurrent breast cancer cases, the population rate of recurrent breast cancer was determined as follows. All individuals with at least three generations of genealogy and a linked death certificate were assigned to a cohort based on biological sex, five-year birth year range, and birth state (Utah or not). Cohort-specific rates of recurrent breast cancer were estimated as the total number of recurrent breast cancer cases in each cohort divided by the total number of individuals with a linked death certificate in each cohort. Each cluster of related, sampled recurrent breast cancer cases (pedigree) was tested for an excess of recurrent breast cancer by comparison of the observed number of recurrent breast cancer cases among the descendants in the pedigree to the expected number. The expected number of cases was estimated by summing the cohort-specific rate of recurrent breast cancer for each descendant who also had a linked death certificate [8].

Whole Exome Sequence (WES) Data Generation and Analysis: Whole exome sequencing was performed at the Huntsman Cancer Institute High Throughput Genomics core on 18 DNA samples extracted from whole blood from recurrent breast cancer cases who were related in five clusters/pedigrees (some individuals were members of more than one of these high-risk pedigrees through different ancestors). DNA libraries were prepared from 1.5 micrograms of DNA using the IDT xGEN Human Exome v2 capture kit. Samples were run on the Illumina NextSeq instrument. Reads were mapped to the human genome GRCh38 reference using BWA-mem for alignment and variants were called using Genome Analysis Toolkit version 4.1.3.0 (GATK) software following Broad Institute Best Practices Guidelines. Exome capture resulted in an average of 95% of target bases being covered by greater than 10× coverage across the exome with an average depth of 196×. Variants occurring outside the exon capture kit intended area of coverage were removed. Variants were annotated with Annovar, which contains predicted pathogenicity scores from 14 in silico functional prediction algorithms.

Candidate-variant association with breast cancer risk: The rare, shared candidate variants identified in the sequencing experiment were analyzed for association with breast cancer risk in a set of 7746 Caucasian breast cancer cases and 1:1 ancestrally matched controls from the UKBiobank’s 488,377 total subjects genotyped on the Illumina OmniExpress SNP array [9]. UKBiobank case and control subjects were matched via principal components (PCs) using ~27 K independent markers that excluded several genomic regions known to adversely affect PC analysis [10]. FLASHPCA2 software was used to generate eigenvectors for control selection [10]. Controls were selected from among 75,447 Caucasian UKBiobank subjects who were female, over age 70 years of age, and had no cancer diagnosis. One control, representing the nearest neighbor based on Euclidean distance of the first two PCs, was selected for each case. A total of 103 outlier cases and controls were removed, leaving 7643 breast cancer cases and controls.

The selected UKBiobank case and control subjects were Imputed to ~40 M SNP markers using the Haplotype Reference Consortium’s (HRC) 67 K background genomes [11]. Beginning with 784,256 observed SNP genotypes, preimputation quality control using PLINK v1.9 software [12] required sample genotyping of >98% (no subjects removed). A total of 353,578 markers were removed by filtering for HWE *p* < 1.0 × 10^−5^, MAF < 0.005, duplicated position in the HRC’s reference genome, or site not included in the HRC’s reference genome. The remaining 430,678 SNPs were converted to human genome B37 forward strand orientation using GenotypeHarmonizer v1.4.20 software [13] and served as the basis for imputation. Imputation was performed with EAGLE v2.3 software for phasing [14] and MINIMAC3 software for imputation [15]. Post-imputation quality control included removing markers with imputation information score (INFO-r2) < 0.7 [16,17,18]. Genomic coordinates were converted to human genome b38 with UCSC liftover tool [19].

Assay of selected candidate variants in additional breast cancer cases: A total of 129 previously sampled, recurrent breast cancer cases were available in the Genetic Epidemiology Biorepository. In addition, sampled breast cancer cases (recurrent status unknown) and connecting relatives from the Biorepository who were members of the five high-risk pedigrees from which a cousin pair was sequenced were selected for Taqman SNP Genotyping (ThermoFisher, Tokyo, Japan) assay. A total of 205 breast cancer cases and relatives were assayed for the selected candidate variants. Due to the limited nature of this study, only 32 candidate variants were selected for assay. Co-author A.W. reviewed the literature to prioritize the candidate genes identified with regard to their putative function in metastasis, with particular attention to the “host” genes that might regulate the body’s ability to keep metastatic recurrence at bay, as well as other plausible contributors.

Variants were submitted to the Thermo Fisher Taqman Functionally Tested SNP Genotyping database based on dbSNP Id. Variants not found to have a functionally tested assay were submitted as custom design assays consisting of 600 base pairs 5 prime and 600 base pairs 3 prime of the variant of interest. Assays were performed by the University of Utah HSC Genomics core.

Protein Structure Simulations: We calculated the 3D structures of the wild type (WT) and mutated (MUT) MDH2 proteins with the I-TASSER structure prediction software using full homology modeling [20,21,22,23]. We used the canonical sequence for MDH2 from UniProt entry P40926. The mutated sequence was obtained by manual replacement of the Valine at position 139 by Isoleucine. The WT I-TASSER predicted structure was compared with experimental structures [24] and with the AlphaFold prediction [25]. All the structures were visualized and analyzed using ChimeraX [26,27], which was also used to calculate and visualize the electrostatic potentials using the default values. The estimated pathogenicity was also estimated using PolyPhen2 [28].

## 3. Results

Recurrent breast cancer high-risk cases and pedigrees: Of the 28,511 breast cancer cases recorded in the UCR with linked genealogy data, 2157 were females with a linked Utah death certificate indicating breast cancer as a cause of death at least 10 years after the primary breast cancer diagnosis (termed *recurrent breast cancer* in this analysis); 129 of these cases had a stored germline DNA sample in the Genetic Epidemiology Biorepository. These 129 recurrent breast cancer cases were related in 59 pedigrees, each pedigree included 2–6 related, sampled recurrent breast cancer cases; 18 of these 59 pedigrees exhibited a significant excess (*p* < 0.05) of recurrent breast cancer cases and were termed high-risk. One of these high-risk pedigrees included five related, sampled, recurrent breast cancer cases that had previously been identified to carry the pathogenic *BRCA1* Q1313X variant. This pedigree was excluded from further study here, but this variant is potentially not only a strong candidate predisposition variant for breast cancer, but specifically for *recurrent* breast cancer. This finding of a known breast cancer predisposition candidate provides solid proof of concept for the power of this study design in predisposition gene identification.

Five of the remaining seventeen sampled, high-risk pedigrees included at least one pair of closely affected relatives (four pairs of cousins, one aunt/niece pair); these pairs were selected for whole exome sequencing in our affected-cousin study design. These five pedigrees included a total of eighteen sampled recurrent breast cancer cases. No sequence data were available for these five pedigrees prior to this study, and they had not been previously screened for *BRCA1* or *BRCA2*. Figure 1, Figure 2, Figure 3, Figure 4 and Figure 5 show the sampled recurrent breast cancer cases (fully shaded) in the five sequenced high-risk pedigrees (two of the sampled cases appeared in more than one high-risk pedigree, through different ancestors). The cases marked with arrows are the related case pairs that were analyzed for variant sharing; the other recurrent breast cancer cases shown also had WES data generated and were analyzed subsequently for segregation of candidate variants. The decade of age at primary breast cancer diagnosis is shown beneath each sequenced recurrent breast cancer case.

Figure 6 summarizes the overall study design and sample sizes for all stages of the study.

Rare, shared variants: Exome sequencing of the 18 affected individuals in five high-risk pedigrees identified a total of ~9200 exon specific (synonymous, nonsynonymous, frameshift, splicing, etc.) variants with MAF < 0.005 in gnomAD 3.0. Removal of synonymous variants and poor-quality variants reduced this to ~4600 variants. A total of 181 rare variants were identified as shared in at least one of the five affected cousin pairs shown in Figure 1, Figure 2, Figure 3, Figure 4 and Figure 5. These candidate recurrent breast cancer candidate variants are listed in Appendix A. No rare variants were observed in more than one pair of affected cousins. Two genes had two different variants identified in cousin pairs from two different pedigrees (*NFATC1* and *ACTL7A*). Five variants were in genes which have been reported to be associated with breast cancer risk in at least 50 manuscripts in PubMed (*DST*, *CHEK2*, *FASN*, *TNN*, and *PMS2*). The variants in *PMS2* and *CHEK2* were classified as pathogenic in ClinVar [29], with reported cancer associations.

Table 1 summarizes the genes in which supporting evidence was identified for rare, shared variants in either UKBiobank analysis or in assay of additional breast cancer cases.

UKBiobank association analysis: A “recurrent breast cancer” phenotype was not identifiable in UKBiobank data, so association of candidate variants with “breast cancer” risk was tested. UKBiobank data was available for 97 of the 181 rare, shared candidate variants; likely due to the low minor allele frequency threshold utilized here (MAF < 0.005). UKBiobank results are summarized in Appendix A. No variants were significantly associated with breast cancer risk in UKBiobank after correction for multiple testing, but variants in *BICD2* (*p* = 0.03) and *TPPP2* (*p* = 0.02, OR = 1.95) were independently significant. Variants in *BICD2* (six cases), *FBN1* (four cases), *CSGALNACT1* (one case), and *ZNF841* (one case) were observed in at least one UKBiobank breast cancer case, but not in any controls. Variants observed in at least four cases and with OR ≥ 2.0 included those in genes *CSMD3* (OR = 6.00; *p* = 0.12), *MUC5AC* (OR = 3.01; *p* = 0.29), and *TMPRSS12* (OR = 2.17; *p* = 0.17).

Assay of selected variants in breast cancer cases: We were only able to assay a subset of the rare, shared candidate variants in additional sampled recurrent breast cancer cases as well as to test for segregation to other sampled *breast cancer affected* relatives in the five high-risk, recurrent breast cancer pedigrees. Of the 32 candidate variants initially selected, seven assays failed (in genes *NFATC1*-both variants, *MR1*, *CHEK2*, *ACTL7A* variant2, *DST*, and *FASN*). Six variants had no additional breast-cancer-affected carriers identified in the pedigrees they were originally found in (in genes *KIF15*, *TMEM192*, *MUC5AC*, *TEP1*, *ZFX*, and *TNN*). Eleven variants identified only one additional breast-cancer-affected carrier in the pedigrees in which they were identified (*CHD5*, *DYSF*, *GZMA*, *AKAP11*, *MED14*, *EPHX1*, *SOX13*, *ACSS2*, *ACTL7A* variant 1, *SLCO1B3*, and *FBN1*). For the variant in *TMEM132C2*, two additional breast-cancer-affected carriers were identified in the pedigree in which they were observed in the recurrent breast cancer case cousin pair. The *KIF1C* variant was identified in three assayed breast cancer cases, but they were not related to the original variant-carrying pair, nor to each other. The *ITGA7* variant was identified in five assayed breast cancer cases, but they were not related to the original variant-carrying cousin pair, nor to each other.

A known pathogenic variant in *CHEK2* as well as a known pathogenic variant in *PMS2* were both observed in the affected cousin pair sequenced and shown in Figure 3 with arrows, but neither variant was observed in the three other sampled recurrent breast cancer cases that were also sequenced in the pedigree shown in Figure 3. Figure 7 shows only the portion of the pedigree with the sequenced affected cousin pair (shown with arrows), and shows other sampled, diagnosed *breast cancer* cases. As seen, the pedigree member parent of each of the *recurrent breast cancer*-affected cousin pair were each diagnosed with breast cancer (one female and one male), as was one of their uncles, and their grandmother (who married into the pedigree shown in Figure 3). The sequenced recurrent breast cancer-affected cousin pair both carried a rare pathogenic variant (PV) in *CHEK2* and a rare PV in *PMS2* (shown with “+”). The *CHEK2* variant assay failed to perform, but the *PMS2* variant was identified in two additional *breast cancer* cases (shown with “+”) by assay. The pedigree shown in Figure 7 is also part of a previously studied breast cancer pedigree that ascends through the male founder shown in the top generation. Sampling and screening of additional breast cancer cases through multiple different ancestors could be pursued to trace segregation of these breast-cancer-associated PVs.

Only three variants were observed not only in the affected cousin pair, but also in the other recurrent breast cancer cases sequenced in the same pedigree (in genes *TENM2*, *CCDC136*, and *MDH2*). All three of these variants were observed to co-segregate with all the sequenced recurrent breast cancer cases in the same pedigree (Figure 1). *TENM2* is on chromosome 5; *CCDC136* (position 128,801,398) and *MDH2* (position 76,058,064) are both on chromosome 7.

Assay of all other sampled breast cancer cases in the pedigree in which the four sequenced *MDH2*- and *TENM2*- and *CCDC136*-variant-carrying recurrent breast cancer cases were originally identified (Figure 1) found one additional breast cancer case carrier of the *TENM2* variant, two additional breast cancer case carriers of the *CCDC136* variant, and seven additional carriers of the *MDH2* variant (one inferred). The *MDH2* variant provided the most evidence for segregation with breast cancer risk; this pedigree is the extension of the pedigree in Figure 1 and is shown in Figure 8.

Figure 8 shows the segregation of the rare candidate *MDH2* variant in the four original sequenced recurrent breast cancer cases (shown with arrows), as well as the additional six breast cancer cases (one is a male breast cancer), and one inferred breast cancer case; decade of age at diagnosis is shown below each case. The founders of this pedigree were born in the early 1800s in Denmark and have over 11,600 descendants represented in the UPDB today. Among all descendants there is a significant excess of breast cancer (83 observed, 60.1 expected, *p* = 0.003), colorectal cancer (44 observed, 29.3 expected, *p* = 0.006), uterine cancer (23 observed, 13.2 expected, *p* = 0.009), melanoma (54 observed, 41.9 expected, *p* = 0.04), pancreas cancer (14 observed, 7.3 expected, *p* = 0.018), thyroid cancer (24 observed, 15.8 expected, *p* = 0.03), and prostate cancer (78 observed, 53.0 expected, 7.7 × 10^−4^). These other cancer cases are not shown, but among the ancestors of variant carriers who are shown in Figure 8, other cancers observed include one pancreas cancer and one prostate cancer; one of the breast cancer case variant carriers was also diagnosed with thyroid cancer. UPDB pedigrees only have linked UCR cancer data from 1966, so cancer cases are typically only identified in the most recent generations.

Predicted functional alteration of *MDH2* V139I: The *MDH2* variant considered here (V139I) is reported as benign in PolyPhen 2 (Appendix A) and ClinVar [29]. These results are not surprising, because of the close structural and electrostatic properties of Valine and Isoleucine. This is also consistent with the comparison of the 3D (predicted) structures of the WT and mutant (MUT) proteins, which are compared in Appendix A. It should be noted that the I-TASSER predicted structure for *MDH2* (WT) compares well with the experimental structure of *MDH2* in the quatrimer complexed with NAD [24] (RMS for 311 pruned pairs of 0.565 Å, for all 314 pairs 0.652 Å) and with the predicted structure of AlphaFold (RMSD between 297 pruned atom pairs is 0.541 Å, across all 338 pairs: 10.851 Å). The graphical comparison of these structures is depicted in Appendix A.

The previous results appear to indicate that there are not key structural differences between the WT and MUT structures of *MDH2* considered here, but further insight can be found when considering the structure of NAD bound to *MDH2* [24], which clearly shows that the V139I mutation is inside of the NAD binding pocket of *MDH2* (see Appendix A). This observation prompted us to examine the electrostatic potential in the region of the V139I mutation and the NAD binding pocket in more detail. The comparison of the coulombic potential in the region of the pocket and the V139I mutation is depicted in Figure 9. It is apparent from the figure that the mutation V139I significantly changes the electrostatic potential, as it appears that there is a much more open and largely bipolar pocket in the WT protein, while the pocket in the MUT protein appears much more narrow and mostly positive.

## 4. Discussion

While there are many informative collections of studied breast cancer families, we are not aware of other existing resources for pedigrees at high risk for *recurrent* breast cancer (recurrence here is defined as death from breast cancer at least 10 years after diagnosis of primary breast cancer). In addition, while most studies can identify clusters of related cases, they do not typically have the necessary population-level data to determine which, if any, of those clusters (or pedigrees) have a significant excess of cases, rather than merely representing a chance cluster of cases. The statewide data combined in the UPDB allowed the identification of pedigrees that exhibit a statistically significant excess of recurrent breast cancer cases over expected rates. The Genetic Epidemiology DNA repository further allowed the identification of stored DNA samples from recurrent breast cancer cases who are members of these high-risk pedigrees; these combined data and resources resulted in the unique study presented here. In this concise sequencing study, a robust method was employed, leveraging previously sampled recurrent breast cancer cases from high-risk pedigrees. The study successfully pinpointed a collection of compelling candidate variants that may contribute to the predisposition of recurrent breast cancer. Since validation of segregation and risk association was based partially on the breast cancer phenotype, rather than on the rarer (and remaining unknown for most cases) phenotype of recurrent breast cancer, conclusions may therefore be limited to breast cancer, rather than recurrent breast cancer, pending further study. These findings warrant further investigation across other breast cancer resources to validate their significance. One variant in *MDH2* showed strong evidence for segregation with recurrent breast cancer, as well as breast cancer, in the extended pedigree in which it was discovered.

The protein structure predictions and the difference in the electrostatic potential observed between the WT and the MUT species in the NAD binding region strongly support the hypothesis that the NAD binding energies are different, leading to alterations in the NAD^+^/NADH ratio in the MUT state compared to the WT state. This suggests higher binding energies required in the MUT state compared to the WT state. This finding may explain why the individuals with the variant are more susceptible to treatment resistance than those with the WT.

*MDH2* encodes mitochondrial malate dehydrogenase 2, which converts malate into oxaloacetate using NAD^+^ ≥ NADH, H+ as part of the citric acid cycle, and therefore regulates NADH [30]. Our molecular modeling studies suggest that the *MDH2* variant found in recurrent breast cancer cases may lead to lower [NADH] and elevated NAD^+^, reducing oxygen consumption and ATP production during the mitochondrial respiration cycle—or, in other words, augmentation of anerobic glycolysis, which has been found to contribute to cancer cell proliferation [31,32]. Other studies have shown that the increased intracellular oxygen concentration, because of altered MDH2 activity, regulates stability of HIF-1α, which is critical for cell survival under hypoxic conditions [30,33]. Altered metabolism caused by mutations in *MDH2* has been implicated in various cancers, including an association with metastasis in pheochromocytomas and paragangliomas [34], and a germline splice site mutation leading to reduced MDH2 activity has been identified in a familial case of paraganglioma [35]. It has been published that *MDH2* is an estrogen-regulated gene and that MDH2 promotes proliferation, migration, and invasion of endometrial cancer cells while suppressing their apoptosis [36]. While *MDH2* mutations have not previously been observed in breast cancer cases, decreased expression of *MDH2* has been observed in triple-negative breast cancer cases compared to HER2-positive breast cancer cases, supporting a role of *MDH2* in breast cancer [37]. Thus, mutations in *MDH2* have multiple functions in modulating cell growth and can inhibit or stimulate cancer growth by deregulating the citric acid cycle.

Pedigree studies based on the UPDB have previously provided the identification of *BRCA1* and *BRCA2* [5,6,7], and more recently have identified additional rare cancer predisposition variants (*GOLM1* [38]; *CELF4* [39]; *FANCM* [40]; *ERF* [41]; *LRBA* [42]; *FGF5* [43]) in similar sets of sampled high-risk pedigrees. The identification of the known pathogenic *PMS2* and *CHEK2* variants as well as the initial *BRCA1 Q1313X*-variant-segregating pedigree found in the present study provide additional validation of this predisposition gene identification approach in high-risk pedigrees. Even with the small number of available sampled pedigrees that were screened here, likely variants were identified to explain some high-risk breast cancer pedigrees. This validation supports the further ascertainment and analysis of Utah high-risk breast cancer pedigrees, as well as the further analysis of the other candidate variants identified here as likely predisposing to recurrent breast cancer.

Strengths of this study include the validation of breast cancer case status with the UCR, the validation of death from breast cancer from Utah death certificates, the existing Utah DNA biorepository, and the extensive genealogy data. The linked data available in the UPDB allowed identification of pedigrees with a significant excess of recurrent breast cancer cases, something most breast cancer family studies are not able to confirm. Limitations of this study include the limited number of sampled high-risk recurrent breast cancer pedigrees and cases analyzed. While the UPDB data resource has been validated in many studies, it must be noted that genealogy data do not always indicate biological relationships. Reliance on a combination of a UCR cancer diagnosis of breast cancer, followed by a death certificate at least 10 years later that noted breast cancer as a cause of death to identify “recurrent breast cancer cases” was not ideal, and it is likely that some recurrent breast cancer cases remained unidentified. Because the Utah founders with available genealogy are primarily of Northern European ancestry, the findings cannot be extended to other populations without independent validation.

## 5. Conclusions

Identification of genetic variants predisposing to recurrent breast cancer could allow early intervention and potential prevention of fatal outcomes for breast cancer for susceptible individuals. The use of protein structure simulations greatly complements the epidemiological studies by generating a plausible pathogenicity hypothesis that can be further explored by more refined, albeit much more computationally demanding, molecular simulation methods. These further investigations include studies of the binding energies using docking and molecular dynamics simulations, and functional studies, which we consider outside the scope of this work, to further understand the pathogenic mechanisms of this mutation.

Large-scale case and pedigree studies strongly suggest that other common cancer predisposition genes may not remain to be identified, and that, rather, most familial cancer predispositions might be the result of many, varied, rare predisposition genes and variants [4]. Whether or not this is the case, studies such as this one, that analyze related affected individuals who are members of high-risk pedigrees, have shown the strong potential to identify many candidate predisposition genes and variants for many different phenotypes. Such studies should be pursued, and the candidate predisposition variants identified are worthy of further exploration.

## Figures and Tables

**Figure 1 cancers-15-05851-f001:**
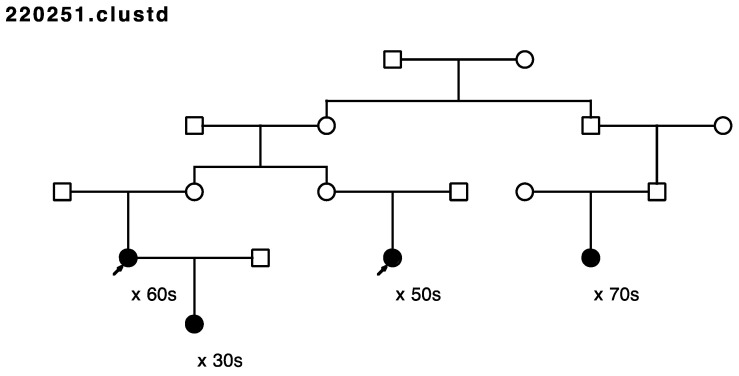
Arrows show the cousin pairs with recurrent breast cancer analyzed for sharing. Variants in genes *MDH2*, *TENM2*, and *CCDC136* were observed in the cousin pair and the two additional recurrent breast cancer cases shown. Thirteen additional previously sampled breast cancer samples were assayed in this pedigree.

**Figure 2 cancers-15-05851-f002:**
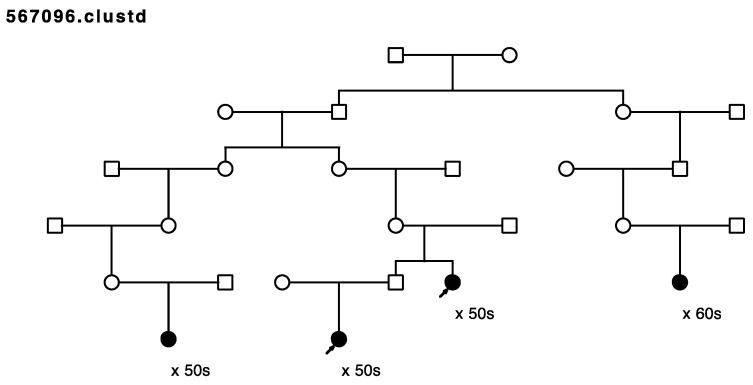
Arrows show the cousin pairs with recurrent breast cancer analyzed for sharing. Two additional related sampled breast cancer cases were assayed in this pedigree.

**Figure 3 cancers-15-05851-f003:**
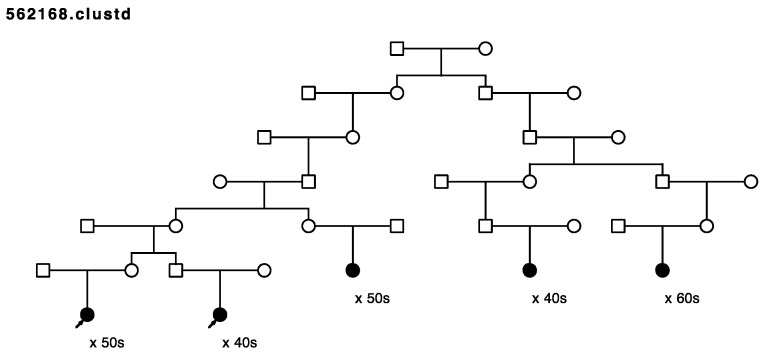
Arrows show the cousin pairs with recurrent breast cancer analyzed for sharing. Three additional related previously sampled breast cancer cases were assayed in this pedigree.

**Figure 4 cancers-15-05851-f004:**
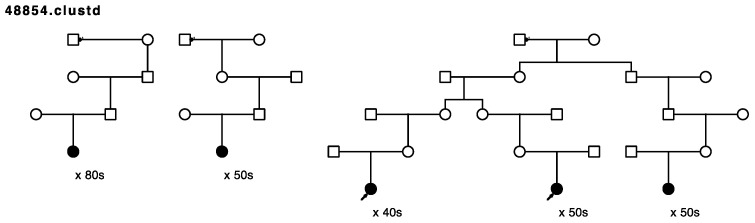
Arrows show the cousin pairs with recurrent breast cancer analyzed for sharing. Eleven additional previously sampled breast cancer cases were assayed in this pedigree. The male founder in the first generation had offspring from three different partners (displayed with a small arrow on the marriage line) with affected descendants in all three lines.

**Figure 5 cancers-15-05851-f005:**
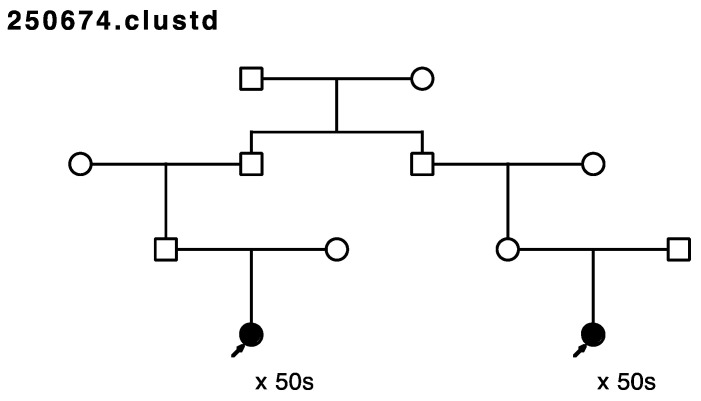
Arrows show the cousin pairs with recurrent breast cancer analyzed for sharing. Nine additional previously sampled breast cancer cases were assayed.

**Figure 6 cancers-15-05851-f006:**
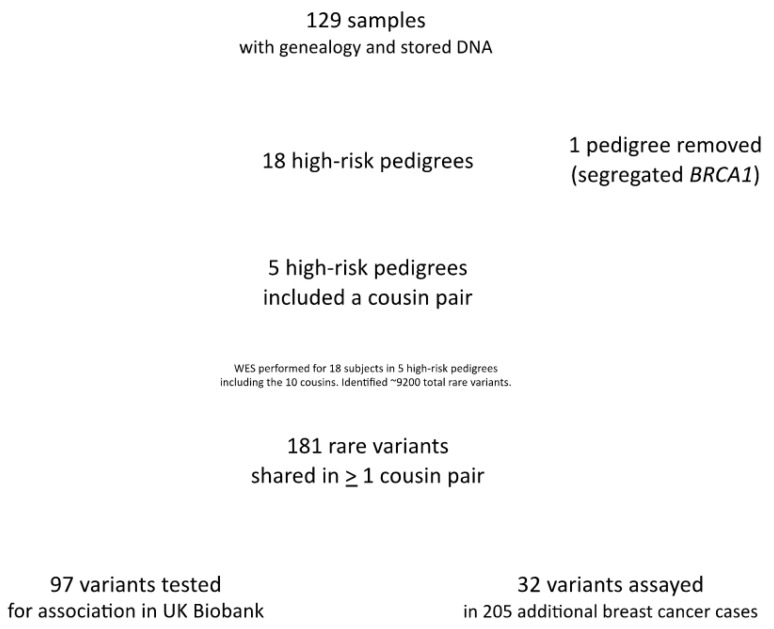
Overall study design and sample size.

**Figure 7 cancers-15-05851-f007:**
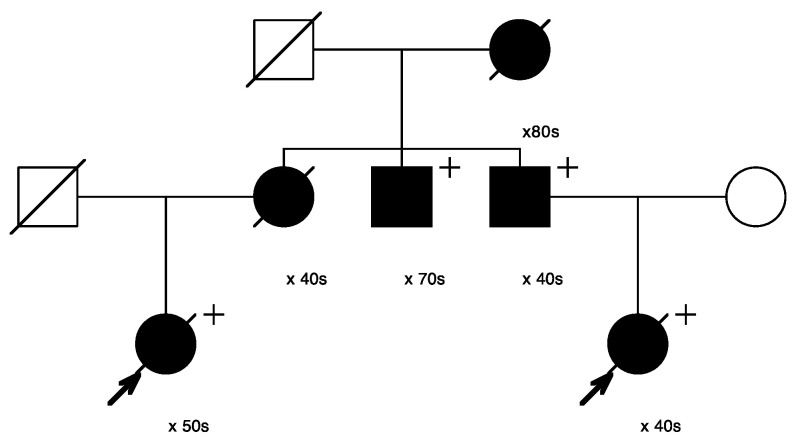
Portion of pedigree in Figure 3 including the cousin pair (arrows) and carriers of the *PMS2* variant (+). The two non-proband breast-cancer-affected cases shown did not have samples available and were not assayed.

**Figure 8 cancers-15-05851-f008:**
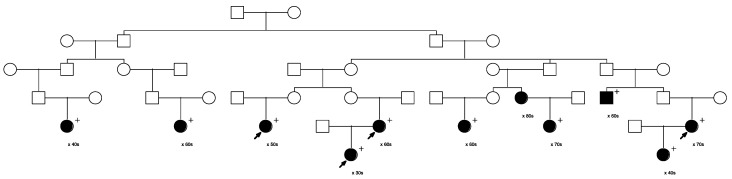
Segregation of the *MDH2* candidate variant in a high-risk recurrent breast cancer pedigree. The arrows show the original recurrent breast cancer cases sequenced, full shading indicates a primary breast cancer diagnosis, and the “+” indicates MDH2 variant carriers.

**Figure 9 cancers-15-05851-f009:**
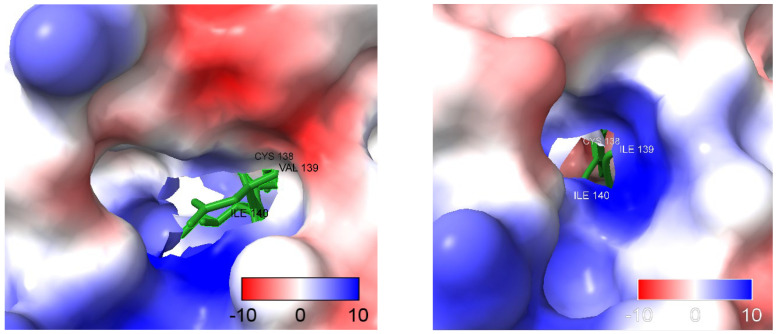
Comparison between the electrostatic potential in the region of the V139I mutation and the NAD binding pocket, where the amino acid sequences MIC**V**IAN and MIC**I**IAN, corresponding to the WT (L) and MUT (R) MDH2, respectively, are depicted in green and the adjacent amino acids to the mutation are labeled.

**Table 1 cancers-15-05851-t001:** Rare shared candidate predisposition variants with supportive evidence in UKBiobank or by assay.

**Genes with Supportive Evidence from UKBiobank Analysis:**			
Gene		UKBiobank Odds Ratio (OR)	UKBiobank *p* value
Independently associated with brca risk:			
*BICD2*		-	0.03
*TPPP2*		1.95	0.02
Observed in at least 1 case, but 0 controls:			
*FBN1* (4 cases)		-	0.125
*CSGALNACT1* (1 case)		-	1.00
*ZNF841* (1 case)		-	1.00
OR > 2.0 and observed in at least 4 brca cases:			
*CSMD3*		6.00	0.12
*MUC5AC*		3.01	0.29
*TMPRSS12*		2.17	0.17
**Genes with Supportive Evidence from Assay of Additional BrCa Cases:**			
	number of Additional *recurrent* brca	number of Additional brca	
Gene	Case carriers in pedigree	Case carriers in pedigree	
*MDH2*	2	7	
*TENM2*	2	1	
*CCDC136*	2	2	
*PMS2*	0	2	

## Data Availability

The data presented in this study are available on request from the corresponding author. The data are not publicly available due to required UPDB approvals for data sharing.

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
