# Peer review of "A Rare Variant in MDH2 (rs111879470) Is Associated with Predisposition to Recurrent Breast Cancer in an Extended High-Risk Pedigree"

_cancers, 2023, doi:10.3390/cancers15245851_

Round 1

Reviewer 1 Report

Comments and Suggestions for Authors

Dear Authors,

The study was conducted reliably and will be of potential interest, especially for readers interested in the hereditary basis of cancer.

My main complaints concern a better graphical presentation of the described results. To improve the manuscript, due to the large number of quoted numbers, all data discussed in the text should refer to appropriate diagrams/tables. It will facilitate reading the manuscript and capturing important information. Please introduce the following changes to the main text:

1.       Flowchart showing the selection of high-risk recurrent breast cancer families and three methodological approaches used (WES, assay of selected candidate variants, and the association analysis using UK Biobank's samples), all with the numbers of families/samples/variants analyzed.

2.       Table/figure (?) listing those shared genetic variants (of the 181) that were confirmed in other samples/somehow significant in additional analyses (assay of selected candidate variants, and the association analysis using UK Biobank's samples).

3.       I think that since the pedigrees of all 5 families are shown, it is important to introduce the names of shared variants into the figures (with "+" or "-") - at least those variants mentioned in the text that were additionally identified in the third or fourth member of a given family.

4.       Please add additional columns to the supplementary table about the analysis of selected variants in other family members or other breast cancer samples, e.g. by adding information on the number of samples tested with/without a given variant.

5. Please add the results generated with the in silico pathogenicity prediction tools for the variant in MDH2 and information on whether the variant is (and how often) present in the population.

6.       It is unclear to me whether the 5 families have previously been tested for mutations in BRCA1/2 or other known high-risk breast cancer genes (and were negative)? Please add appropriate information to the text.

7.       In the "Abstract", please add information that the identified rare variant in MDH2 affected only one family.

8.       Has the MDH2 variant been tested or is it possible to test it in family members (from Figure 7) not affected by cancer?

9.       Author's comment is visible in Figure S4, please remove it.

Best regards.

Author Response

Thank you for the very helpful reviews.  We have responded as indicated below and find the manuscript much improved!

Response to Reviewer 1:

The study was conducted reliably and will be of potential interest, especially for readers interested in the hereditary basis of cancer. My main complaints concern a better graphical presentation of the described results. To improve the manuscript, due to the large number of quoted numbers, all data discussed in the text should refer to appropriate diagrams/tables. It will facilitate reading the manuscript and capturing important information. Please introduce the following changes to the main text:

  1. Flowchart showing the selection of high-risk recurrent breast cancer families and three methodological approaches used (WES, assay of selected candidate variants, and the association analysis using UK Biobank's samples), all with the numbers of families/samples/variants analyzed.

We inserted a flowchart (Figure 6) to RESULTS to display the study design and sample sizes.

  1. Table/figure (?) listing those shared genetic variants (of the 181) that were confirmed in other samples/somehow significant in additional analyses (assay of selected candidate variants, and the association analysis using UK Biobank's samples).

 We have added Table 1 to RESULTS summarizing the variants with supportive evidence identified in UKBiobank and in the assay of additional breast cancer cases.

  1. I think that since the pedigrees of all 5 families are shown, it is important to introduce the names of shared variants into the figures (with "+" or "-") - at least those variants mentioned in the text that were additionally identified in the third or fourth member of a given family.

In RESULTS we state that only for variants in the 3 genes (MDH2, TENM2 and CCDC136) were the rare variants that were found in the cousin pair also present in any of the other recurrent breast cancer cases shown (and in these 3 cases the variants were seen in all the cases shown).  We have added this detail to the legend of Figure 1, the pedigree in which this was observed.

We have additionally added to the Figure 6 legend concerning the additional breast cancer case carriers of the PMS2 variant.

  1. Please add additional columns to the supplementary table about the analysis of selected variants in other family members or other breast cancer samples, e.g. by adding information on the number of samples tested with/without a given variant.

Since all 205 additional previously sampled breast cancer cases and relatives were assayed for the subset of candidates, and few rare variant carriers were identified, rather than adding a significant amount of data and columns to the supplementary table, we have instead noted the number of additional previously sampled breast cancer cases on the pedigree figures. Specific counts of selected candidate carriers are noted in the text in RESULTS.

  1. Please add the results generated with the in silicopathogenicity prediction tools for the variant in MDH2and information on whether the variant is (and how often) present in the population.

We have added columns for the GnomAD frequencies and the in silico predictions to the Supplemental Table for all candidate variants.

  1. It is unclear to me whether the 5 families have previously been tested for mutations in BRCA1/2 or other known high-risk breast cancer genes (and were negative)? Please add appropriate information to the text.

No sequence data was available for these five pedigrees prior to this study, and they had not been previously screened for BRCA1 or BRCA2. A sentence has been added in RESULTS explaining this.

  1. In the "Abstract", please add information that the identified rare variant in MDH2affected only one family.

The information has been added to the ABSTRACT.

  1. Has the MDH2variant been tested or is it possible to test it in family members (from Figure 7) not affected by cancer?

There are additional previously sampled individuals in this pedigree. this additional evaluation step for this variant, while not yet performed, should be one focus of future investigations.

  1. Author's comment is visible in Figure S4, please remove it.

The author’s comment has been fixed in Figure S4.

Reviewer 2 Report

Comments and Suggestions for Authors

This manuscript addresses the possibility that the unusual clustering of recurrent breast cancer disease in some families could be due to heritable genetic factors, which predispose to both breast cancer risk and increased risk of disease recurrence. A unique historical biobank of based on Utah Population Database was used to test this hypothesis. Based on definition of disease recurrence as reported deaths due to breast cancer that occurred at least 10 years, the authors identified 129 DNA samples that belonged to individuals classified as having had recurrent breast cancer and that are related in 59 pedigrees, where 18 of the 59 pedigrees exhibited an excess of recurrent breast cancer cases.  Whole exome sequencing data from such selected samples from high-risk breast cancer cases identified sharing of 181 rare alleles in relatives (cousin relationship). The authors then investigated a UKBiobank resource with goal of identifying associations of any of 181 candidate variants with breast cancer risk but identified data for 97/181 variants where none were found significantly associated with risk though there were some suggestive findings with three rare variants. Then 32 of the 181 candidate variants were genetically analyses in DNA from Utah breast cancer resource including other member members of selected pedigrees. In these analyses an MDH2 V139I variant was identified to segregate with recurrent breast cancer cases in an extended pedigree.

The strategy used to identify the MDH2 V139I variant from the Utah resources is interesting. The number of cases identified to harbour this variant in an extended family is compelling evidence that it could be associated with risk to breast cancer or risk to recurrent breast cancer disease. However, the main issue with this study is the presentation of the data arising from the genetic analyses of various resources (Utah and UK Biobanks) that is confusing and hard to follow in the results section. It is not clear to me that UK Biobank resource played any role in identifying the MDH2 V139I variant or in streamlining the 181 candidate variants identified in Utah resource. Perhaps a flow chart summarizing the findings from each biobank would have been useful.

Other comments:

1)     One BRCA1:Q1313X variant carrier pedigree was removed from the pool of 18 families exhibiting an excess of recurrent breast cancer cases, when retaining it in the study might have been interesting from a proof-of-concept aspect. 

2)     Clarify the distribution of the 181 candidate variants identified with respect to the 18 affected individuals in 5 high-risk pedigrees - perhaps indicate this in Supp Table 1.  Co-occurring variants are an interesting observation especially if identified in known cancer predisposition genes, such as the cousin pair both found to carry variants in PMS2 (MAF = 0) and CHEK2 (MAF 0.0001), genes which are known to confer risk to hereditary cancer. These findings are intriguing given their MAFs are considerably lower (if detectable) in control gnomAD population database as compared to variant found in MDH2 (MAF 0.0036) (see also note below in item 5). 

3)     While there may be only two examples of genes (NFATC1 and ACTL7A) harbouring different variants in different cousin pairs, this result is also interesting leading to question if there are variants in different genes that are involved in similar molecular pathways across the Utah cases? For example, known and proposed breast cancer predisposing genes share common functionalities in their involvement in homologous DNA repair pathways. 

4)     The rationale for using the UKBiobank is clear even though it is acknowledged that their definition of recurrent breast cancer disease could not be applied to this resource.  However, what is not clear why only investigate the 181 candidate variants (identified in the Utah families) rather than any rare variant occurring in any of the genes harbouring them in this resource? A carrier of pathogenic variants in some of the known breast cancer predisposition genes (ie BRCA1) could harbour any one of hundreds deleterious genetic anomalies. Is such data not available from the UKBiobank? 

5)     The fact that only 97 of 181 candidate variants were identified in the UK Biobank and that none of these (including the candidate variant in MDH2) were found significantly associated with breast cancer risk limits the usefulness of findings of this resource in the study design. The authors comment that this may be due to low MAF threshold (<0.005) utilized “here” (lines 208-209). Elaborate on MAF cut-offs used with respect to identification of new genetic markers of risk based on expected model of risk of proposed candidate alleles i.e., rare alleles which have moderate-high effect size versus rare or more common alleles with lower effect size as the authors do not include a discussion on how this would impact their study design. 

6)     A limiting factor of this study is the small number of candidates investigated genetically in breast cancer cases from the Utah resource and that the rationale for selecting any of these 32 from 181 candidate variants was not described in the study design.  Given that that it was not feasible to perform genetic assays on all 181 variants, it would be important to understand the rationale for investigating how the 32 variants were selected for these experiments.

7)     The high allele frequency of the MDH2 V139I variant is the general population is higher than one would expect for a specific rare disease associated allele (i.e. akin to a deleterious variant in BRCA1) and this should be addressed. Also, the various cancer phenotypes exemplified in the carrier family should be elaborated upon including their carrier status for this variant. Could the cases having other types of cancer also be shown in Figure 7? 

8)     A more compelling argument that MDH2 variant is associated with recurrent breast cancer would be the identification of another family with recurrent breast cancer or recurrent breast cancer cases harbouring in this variant or other potentially damaging variants in this same gene, especially considering the allele frequency of MDH2 V139I variant in the general population.

Author Response

Thank you for the very helpful reviews.  We have responded as indicated below and find the manuscript much improved!

Response to Reviewer 2

… However, the main issue with this study is the presentation of the data arising from the genetic analyses of various resources (Utah and UK Biobanks) that is confusing and hard to follow in the results section. It is not clear to me that UK Biobank resource played any role in identifying the MDH2 V139I variant or in streamlining the 181 candidate variants identified in Utah resource. Perhaps a flow chart summarizing the findings from each biobank would have been useful.

It is the case that the candidate variants assayed for validation purposes were selected before the UKBiobank association analysis was completed. These association results did not independently validate MDH2, but did add some support for some candidates, which is discussed.

1)     One BRCA1:Q1313X variant carrier pedigree was removed from the pool of 18 families exhibiting an excess of recurrent breast cancer cases, when retaining it in the study might have been interesting from a proof-of-concept aspect. 

We agree that this finding provides solid proof of concept for our study design and findings and have added a comment in RESULTS.

2)     Clarify the distribution of the 181 candidate variants identified with respect to the 18 affected individuals in 5 high-risk pedigrees - perhaps indicate this in Supp Table 1.  Co-occurring variants are an interesting observation especially if identified in known cancer predisposition genes, such as the cousin pair both found to carry variants in PMS2 (MAF = 0) and CHEK2 (MAF 0.0001), genes which are known to confer risk to hereditary cancer. These findings are intriguing given their MAFs are considerably lower (if detectable) in control gnomAD population database as compared to variant found in MDH2 (MAF 0.0036) (see also note below in item 5). 

We have amended Supplementary Table 1 to include a column indicating in which pedigree the candidate variant was identified as shared in the cousin pair.

3)     While there may be only two examples of genes (NFATC1 and ACTL7A) harbouring different variants in different cousin pairs, this result is also interesting leading to question if there are variants in different genes that are involved in similar molecular pathways across the Utah cases? For example, known and proposed breast cancer predisposing genes share common functionalities in their involvement in homologous DNA repair pathways. 

73 of the candidate variants identified here belong to pathways associated with breast cancer; these are identified in a column of Supplemental Table 1.  There are many molecular pathways representing the candidates identified and we feel that their exploration can be a part of future studies.

4)     The rationale for using the UKBiobank is clear even though it is acknowledged that their definition of recurrent breast cancer disease could not be applied to this resource.  However, what is not clear why only investigate the 181 candidate variants (identified in the Utah families) rather than any rare variant occurring in any of the genes harbouring them in this resource? A carrier of pathogenic variants in some of the known breast cancer predisposition genes (ie BRCA1) could harbour any one of hundreds deleterious genetic anomalies. Is such data not available from the UKBiobank? 

In our many other investigations of rare cancer predisposition variants we typically identify only approximately half of the variants in the UKBiobank dataset to which we have access; we assume this is due to the low MAF threshold we use.  Genetic sequence data in UKBiobank is now becoming available; in future investigations, this will allow identification/consideration of all variants as well as identification of co-occurring variants in known genes.

5)     The fact that only 97 of 181 candidate variants were identified in the UK Biobank and that none of these (including the candidate variant in MDH2) were found significantly associated with breast cancer risk limits the usefulness of findings of this resource in the study design. The authors comment that this may be due to low MAF threshold (<0.005) utilized “here” (lines 208-209).

We agree that the findings of the UKBiobank analysis were not particularly useful in this project; nevertheless, we felt it important to share what we did and what was observed. For this reason we list all of our identified rare variants that had any suggestive evidence in UKBiobank.  Again, our intent is to suggest strong candidate variants that can be explored in other studies with more data.

Elaborate on MAF cut-offs used with respect to identification of new genetic markers of risk based on expected model of risk of proposed candidate alleles i.e., rare alleles which have moderate-high effect size versus rare or more common alleles with lower effect size as the authors do not include a discussion on how this would impact their study design. 

This small study was designed to show the power and efficiency of the high-risk pedigree approach utilized. Therefore, by design we are searching for, and powered to identify, low frequency variants. As the reviewer suggests, the frequency threshold could be increased to allow more candidate variants to be identified, but this modification would increase the noise.  The frequency cutoff for larger (perhaps population-wide) studies of high-risk pedigrees should be a consideration, but in this study it provided us with a manageable list of candidate variants to consider.

6)     A limiting factor of this study is the small number of candidates investigated genetically in breast cancer cases from the Utah resource and that the rationale for selecting any of these 32 from 181 candidate variants was not described in the study design.  Given that that it was not feasible to perform genetic assays on all 181 variants, it would be important to understand the rationale for investigating how the 32 variants were selected for these experiments.

Co-author Alana Welm, PhD is focused on the problem of breast cancer metastasis using in vivo modeling of mouse and human breast cancers. Since we had limited funds, Dr. Welm was asked to consider the 181 variants and select a subset for assay. We have added this detail in Methods.

7)     The high allele frequency of the MDH2 V139I variant is the general population is higher than one would expect for a specific rare disease associated allele (i.e. akin to a deleterious variant in BRCA1) and this should be addressed.   

While the frequency might be higher than expected, there is quite a lot of variability for allele frequencies across populations; the MDH2 pedigree observed is a strong argument supporting this variant. We have added columns for the GnomAD frequencies to the Supplemental Table for readers.

Also, the various cancer phenotypes exemplified in the carrier family should be elaborated upon including their carrier status for this variant. Could the cases having other types of cancer also be shown in Figure 7? 

All of the cancer phenotypes observed in individuals with MDH2 variants are mentioned in the text.  In order to protect confidentiality and limit identifiability we are prohibited from showing other cancers in Figure 7 and we limit diagnosis age data to 10 year ranges.

8)     A more compelling argument that MDH2 variant is associated with recurrent breast cancer would be the identification of another family with recurrent breast cancer or recurrent breast cancer cases harbouring in this variant or other potentially damaging variants in this same gene, especially considering the allele frequency of MDH2 V139I variant in the general population.

We completely agree with the reviewer.  We hope this manuscript will encourage investigations into independent resources that might add supportive data for MDH2 or other of our candidate variants.

Round 2

Reviewer 2 Report

Comments and Suggestions for Authors

Interesting study and findings that would be interest to other experts in the field of hereditary breast cancer. The comments and alterations to manuscripts adequately addressed my concerns.